# Towards Minimal Targeted Updates of Language Models with Targeted Negative Training

**Lily H. Zhang**                                                              *lily.h.zhang@nyu.edu*
*New York University*

**Rajesh Ranganath**                                                          *rajeshr@cims.nyu.edu*
*New York University*

**Arya Tafvizi**                                                              *aryatafvizi@google.com*
*Google*

**Reviewed on OpenReview:** *https://openreview.net/forum?id=lrZ2yiqOS2*

## Abstract

Generative models of language exhibit impressive capabilities but still place non-negligible probability mass over undesirable outputs. In this work, we address the task of updating a model to avoid unwanted outputs while minimally changing model behavior otherwise, a challenge we refer to as a minimal targeted update. We first formalize the notion of a minimal targeted update and propose a method to achieve such updates using negative examples from a model's generations. Our proposed Targeted Negative Training (TNT) results in updates that keep the new distribution close to the original, unlike existing losses for negative signal which push down probability but do not control what the updated distribution will be. In experiments, we demonstrate that TNT yields a better trade-off between reducing unwanted behavior and maintaining model generation behavior than baselines, paving the way towards a modeling paradigm based on iterative training updates that constrain models from generating undesirable outputs while preserving their impressive capabilities.

## 1 Introduction

Despite their impressive achievements, language models still output undesirable text. Examples include hallucinations (Maynez et al., 2020; Martindale et al., 2019; Raunak et al., 2021; Ji et al., 2022; Huang et al., 2021), toxic language (Gehman et al., 2020), and context-specific forms of unwanted outputs, from improper style (e.g. informal language in contexts where formality is expected) to inappropriate content (e.g. advanced topics in applications for children).

In recent years, various strategies have been proposed to control the generations of an existing language model by changing the sampling process during inference time, e.g. via a guided decoding strategy based on rules (Paulus et al., 2018; Hokamp & Liu, 2017), auxiliary models (Dathathri et al., 2019; Krause et al., 2021; Yang & Klein, 2021; Liu et al., 2021), or prompt design (Brown et al., 2020).[1] Such techniques, however, add latency or complexity to the prediction, as they push all desired model changes to inference time; moreover, the costs to the prediction pipeline only increase as the list of changes grows, whether that is maintaining a codebase of decoding rules and specialized prompts, or running multiple auxiliary models to guide the original model's decoding. As language models become more ubiquitous across product stacks, their ease of use and speed during prediction will be increasingly important, and the strategy of pushing all model changes to inference time will become increasingly impractical.

In this work, we instead consider training-time strategies for improving a model's generations. The most naive way to address problems with an existing model is to train a new one on modified data or with an

---

[1]For a given starting input $\mathbf{c}$, prompt design strategies sample from $p(\mathbf{x} \mid \mathbf{c}'), \mathbf{c}' \neq \mathbf{c}$ instead of $p(\mathbf{x} \mid \mathbf{c})$.

alternative training strategy. However, retraining to address one problem can result in a model that exhibits new problems. As an example, Welbl et al. (2021) show that training on data filtered to be less toxic hurts model perplexity, disproportionately so for text associated with minority groups. The same can be said about finetuning, which does not start the modeling process from scratch but can still result in models that are substantially different from their base versions, e.g., due to catastrophic forgetting (McCloskey & Cohen, 1989; Kirkpatrick et al., 2016; Luo et al., 2023). Thus, finetuning can also suffer from the same problems as retraining from scratch, namely that new problems emerge in the endeavor to address existing ones; for instance, Xu et al. (2021) find that finetuning an existing language model on detoxified data also hurts model perplexity disproportionately on text with minority group mentions, and the degradation increases the longer the finetuning. *Is there a way to adapt finetuning to make more targeted changes to a model?*

In this work, we propose a finetuning strategy for minimal targeted updates to an existing autoregressive generative model. Minimal targeted updates constrain an existing model to avoid certain behavior while keeping the resulting model close to the original. The proposed approach in this work, called Targeted Negative Training (TNT), uses only the original model and annotations of its generations to target a new model that is a minimal targeted update of the original. TNT does not affect inference time, unlike decoding-time procedures for controllable generation, and its focus on negative examples allows it to target changes (i.e., avoiding outputs) that would be much more difficult for methods which focus on other data types, from "positive" demonstrations to preference data.

We first discuss the challenge of constraining model generations and why existing common finetuning strategies fall short (Section 2). Then we propose TNT, a finetuning solution for avoiding undesirable outputs via a minimal targeted change (Section 3). We next compare TNT to other related work in the literature (Section 4) and show in experiments that TNT enables more precise control than baselines over the trade-off between reducing unwanted behavior and maintaining existing model behavior (Section 5). Code for TNT can be found at `https://github.com/google/t5patches`.

## 2 The Challenge of Constraining Model Generations

In this section, we motivate and define a minimal targeted update. First, we discuss the limitations of coarse data filtering, motivating the use of token-level annotations (Section 2.1). Then, we explain how existing losses for negative signal fail to govern where probability mass should be redispersed, motivating the need for objectives that not only push down probability mass but also control what the resulting distribution should look like (Section 2.2). Then, we define the solution to a minimal targeted update (Section 2.3).

### 2.1 Data Filtering is a Coarse Solution

In general, a text sequence is undesirable not because every single token in the text is bad, but rather because some subset of the text conveys unwanted content. Data filtering however removes not only bad content but all of the text that co-occurs with bad content. As a result, language and concepts that happen to be correlated with bad behavior become under-represented in the finetuning distribution. The toxicity examples in the introduction provide one such example, and our own experiments on reducing hallucination (details in Section 5), we also find that retraining or finetuning on filtered data can significantly change the generation behavior of a model beyond the change of interest (results in Appendix C). Methods which build on finetuning with filtered data (e.g., Ilharco et al. (2023)) are also susceptible to this same issue.

Finetuning on token-level annotations can ameliorate the above issue by enabling a training loss that treats unwanted tokens differently from others. Such an approach allows all acceptable text to contribute to model training, even text that is correlated with unwanted text. The effort required to collect such token-level annotations can be expensive but in some cases may be comparable to that of collecting sequence-level annotations—for instance, labeling an overall sequence with "has hallucination" or "has offensive language" generally requires identifying the hallucination or offensive language itself. In fact, Wu et al. (2023) find that annotation time is similar for fine-grained and sequence-level labels in a long-form QA task, but finetuning a model on the former yields substantial performance benefits over finetuning on the latter.

Next, we consider existing losses that operate on token-level negative signal.

## 2.2 Existing Negative Losses do not Control the Resulting Distribution

Here we show that existing objectives that take into account negative signal are insufficient to enforce targeted updates. Given a distribution of negative (i.e. unwanted) examples $p^{\text{neg}}$, negative likelihood (NL) (He & Glass, 2020) negates the log-likelihood objective for this distribution: $\mathcal{L}_{\text{NL}}(\theta) = -\mathbb{E}_{p^{\text{neg}}}[-\log p_\theta(x)]$. Unlikelihood (UL) (Welleck et al., 2020) instead maximizes $\log(1 - p(x))$ over the distribution of negatives: $\mathcal{L}_{\text{UL}}(\theta) = -\mathbb{E}_{p^{\text{neg}}}[\log(1 - p_\theta(x))]$. These token-level losses for negative signal are typically combined with log-likelihood (log likelihood (LL)) on acceptable tokens as positive signal: for instance, for UL, we have $\mathcal{L}(\theta) = -\sum_t \left( \mathbf{1}[\mathbf{x}_t \in \text{supp}(p^{\text{neg}}_{\mathbf{c}, \mathbf{x}_{<t}})] \log[1 - p_\theta(\mathbf{x}_t | \mathbf{c}, \mathbf{x}_{<t})] + \mathbf{1}[\mathbf{x}_t \notin \text{supp}(p^{\text{neg}}_{\mathbf{c}, \mathbf{x}_{<t}})] \log p_\theta(\mathbf{x}_t | \mathbf{c}, \mathbf{x}_{<t}) \right)$. We first consider the negative token-level losses by themselves and then objectives that combine these negative losses with log-likelihood on positive tokens (i.e., NL + LL and UL + LL).

First, both NL and UL alone are optimized when all negative examples have zero probability under the model distribution. However, because both losses are defined as expectations with respect to the distribution of negatives $p^{\text{neg}}$, neither NL or UL account for how mass is dispersed outside of the negative tokens. In other words, both NL and UL reduce the probability of the target negative tokens but do not control what the new probability distribution will look like at that token index, since tokens $x \notin p^{\text{neg}}$ do not factor into the expectation for either loss. For instance, under these objectives, a distribution that places all of its mass on one particular element outside the support of negative examples is indistinguishable from a distribution that spreads its probability mass arbitrarily across all non-negative examples. In other words, these negative losses push down probability mass but do not specify how probability mass should be redistributed.

Even when we combine these negative losses with log-likelihood over positive tokens, the overall objectives still do not specify the solution an update should target in the context of finite data. For NL + LL, the negative likelihood is unbounded (lowest value is $-\infty$) and thus can outweigh the log likelihood components of the loss (lowest value is 0) that encourage pushing up probability over acceptable tokens. Unlike NL, UL is bounded (lowest value is 0), but without specifying what the token distributions should be for indices with negative examples, the resulting UL + LL loss does not sufficiently control the target of the update. In fact, we find that utilizing these objectives increases the prevalence of disfluencies in generated sequences relative to the original model; for instance, using NL on negative tokens and LL on positive tokens increases the frequency of word repeats by 17x and 38x on the datasets we test, while using UL introduces a ?? disfluency to 1.1% and 5.4% of the generations respectively (see Table 1 for details). These occurrences highlight the need to define the solution to a minimal targeted update for negative examples, which we do next.[2]

## 2.3 Defining the Solution to a Minimal Targeted Update

While losses such as negative likelihood and unlikelihood do not define where probability mass should be dispersed in a negative update, here we define the solution to a minimal targeted update: Given an original distribution $p^o(\mathbf{x})$, a minimal targeted update results in a new distribution $p^{\text{new}}(\mathbf{x})$ that is closest to $p^o(\mathbf{x})$ in reverse KL-divergence while meeting a desired criterion, namely to avoid certain unwanted outputs. The choice of reverse KL-divergence is a natural one in this setting since the goal is to constrain the support of the original distribution, and the forward KL-divergence is infinite when the support of the new distribution is a strict subset of the original.

Let the distribution of unwanted elements be $p^{\text{neg}}$. The model should not output negative examples, i.e., $\mathbf{x} \in \text{supp}(p^{\text{neg}})$. Let $\mathcal{P}_k$ denote the set of distributions $p_k$ which satisfy the criterion $\forall \mathbf{x} \in \text{supp}(p^{\text{neg}}), \ p^k(\mathbf{x}) = 0$. Then, we define result under a minimal targeted change as

$$p^{\text{new}} = \min_{p^k \in \mathcal{P}_k} \text{KL}(p^k || p^o). \tag{1}$$

The distribution $p^{\text{new}}$ is also known as the information projection of the original distribution $p^o$ onto $\mathcal{P}_k$ and is guaranteed to be unique given $\mathcal{P}^k$ is a closed, convex set (Csiszár & Shields, 2004). Its solution is

$$p^{\text{new}}(\mathbf{x}) \propto p^o(\mathbf{x}) \mathbf{1}[\mathbf{x} \notin \text{supp}(p^{\text{neg}})]. \tag{2}$$

---

[2]For clarification, we note that the original implementations of these losses paired them with positive signal not just on other token indices, but also on the same index as the negative signal. This is not available in the context of an update with negative examples, as it would require providing corrections for the unwanted tokens.

See Appendix A for details. Equation (2) is a special case of the information projection for pointwise constraints (i.e., constraints that are applied to every element in the distribution) and offers a simple mathematical form for a minimal targeted change. Namely, $p^{\text{new}}$ is the distribution that would be obtained by pushing the probability of the negative examples down to zero under the original distribution and renormalizing. This solution encompasses a wide range of applications, as the set of negative examples $\text{supp}(p^{\text{neg}})$ can generalize to any set one wishes to avoid, from factual inaccuracies to offensive language to text in a certain style.

**Relationship to Conditioning.** The distribution in Equation (2) is equivalently the distribution of the original model conditioned on the criterion $\mathbf{x} \notin \text{supp}(p^{\text{neg}})$:

$$p^o(\mathbf{x} \,|\, \mathbf{x} \notin \text{supp}(p^{\text{neg}})) \propto p^o(\mathbf{x})p^o(\mathbf{x} \notin \text{supp}(p^{\text{neg}}) \,|\, \mathbf{x}) = p^o(\mathbf{x})\mathbf{1}[\mathbf{x} \notin \text{supp}(p^{\text{neg}})]. \tag{3}$$

This correspondence implies that methods which use Bayes' Rule to condition on a constraint (Krause et al., 2021; Yang & Klein, 2021) are performing a minimal targeted update during inference.[3] Conversely, any minimal targeted update can also be viewed via the lens of conditioning. We now define TNT to target this solution directly.

## 3 Targeted Negative Training

TNT seeks to approximate the desired generator $p^{\text{new}}$ (Equation (2)) directly via a model $p_\theta$, rather than change the sampling procedure for the original model $p^o$ during inference.

To encourage targeted probability removal, TNT uses the following insight: a single forward pass through a language model provides a single sample from a high-dimensional distribution over sequences, but the same forward pass provides a fully specified distribution over tokens for every prefix that makes up the sequence. In other words, while one can only estimate $\mathbb{E}_{\mathbf{x} \sim p^{\text{new}}}[\log p_\theta(\mathbf{x})]$ via a Monte Carlo approximation of the high-dimensional distribution $p^{\text{new}}$, it is possible to analytically compute the analogous expression for the constituent token distributions $p^{\text{new}}_{\mathbf{c},\mathbf{x}_{<t}}$ defined by input $\mathbf{c}$ and output prefix $\mathbf{x}_{<t}$. $p^{\text{new}}_{\mathbf{c},\mathbf{x}_{<t}}$ is obtained by taking original token distribution $p^o_{\mathbf{c},\mathbf{x}_{<t}}$, removing probability mass from all elements in the negative token distribution $p^{\text{neg}}_{\mathbf{c},\mathbf{x}_{<t}}$, and renormalizing.

TNT simply minimizes a divergence between the model distribution $p_{\theta,\mathbf{c},\mathbf{x}_{<t}}$ and desired distribution $p^{\text{new}}_{\mathbf{c},\mathbf{x}_{<t}}$ for every token distribution $p_{\mathbf{c},\mathbf{x}_{<t}}$ encountered in the training set, given examples from $p^{\text{neg}}_{\mathbf{c},\mathbf{x}_{<t}}$. In this work negative examples come from annotations of the original model's generations, e.g. spans that are labeled bad, but they can also be specified up front without referencing model generations, as has been done previously to reduce repetition and contradictions in neural text generation (Welleck et al., 2020; Li et al., 2020), or given by external classifiers that operate on text prefixes (Yang & Klein, 2021). When there are no negative examples for a given $p_{\mathbf{c},\mathbf{x}_{<t}}$, then $p^{\text{new}}_{\mathbf{c},\mathbf{x}_{<t}} = p^o_{\mathbf{c},\mathbf{x}_{<t}}$.

Because a language model can be defined by its constituent distributions $p_{\mathbf{c},\mathbf{x}_{<t}}$, if all distributions $p_{\mathbf{c},\mathbf{x}_{<t}}$ match the desired $p^{\text{new}}_{\mathbf{c},\mathbf{x}_{<t}}$, then the overall model matches $p^{\text{new}}$. However, because it is computationally impractical to enumerate every constituent distribution $p_{\mathbf{c},\mathbf{x}_{<t}}$, we instead opt to constrain the distributions that are more likely to be relevant in the generations for a given task, as approximated by the original model's generations. Thus, given a task specified by a distribution over input queries $p(\mathbf{c})$, TNT optimizes for a sequence-to-sequence model that approximates $p^{\text{new}}_{\mathbf{c},\mathbf{x}_{<t}}$ as well as possible on average, where the average is defined by the original model's generation process. In other words, the distributions $p_{\mathbf{c},\mathbf{x}_{<t}}$ that are more likely in decoding under the original model are also more likely for training the new model. For a given choice of divergences $\text{D}_p$ and $\text{D}_n$, TNT's objective is

$$\begin{aligned}
\mathcal{L}(\theta) = \mathbb{E}_{\mathbf{c} \sim p(\mathbf{c})}\mathbb{E}_{\mathbf{x} \sim p^o_{\mathbf{c}}(\mathbf{x})}\Big[ &\sum_{t=1}^{\text{len}(\mathbf{x})} \mathbf{1}[\mathbf{x}_t \in p^{\text{neg}}_{\mathbf{c},\mathbf{x}_{<t}}]\text{D}_n\big(p^{\text{new}}_{\mathbf{c},\mathbf{x}_{<t}}(\mathbf{x}_t)||p_{\theta,\mathbf{c},\mathbf{x}_{<t}}(\mathbf{x}_t)\big) \\
&+\mathbf{1}[\mathbf{x}_t \notin p^{\text{neg}}_{\mathbf{c},\mathbf{x}_{<t}}]\text{D}_p\big(p^{\text{new}}_{\mathbf{c},\mathbf{x}_{<t}}(\mathbf{x}_t)||p_{\theta,\mathbf{c},\mathbf{x}_{<t}}(\mathbf{x}_t)\big)\Big].
\end{aligned} \tag{4}$$

---

[3]The aforementioned cases do not force the constraint to be hard, i.e. $p(\mathbf{x} \notin \text{supp}(p^{\text{neg}}) \,|\, \mathbf{x})$ can lie between 0 and 1, but in practice, the types of control desired for generation implies hard constraints that the text either meets or does not, i.e. $p(\mathbf{x} \notin \text{supp}(p^{\text{neg}}) \,|\, \mathbf{x}) \in \{0, 1\}$.

To implement this loss, we generate outputs from the original model and annotate them for undesirable text. Then, we obtain $p^o_{\mathbf{c},\mathbf{x}_{<t}}$ and $p_{\theta,\mathbf{c},\mathbf{x}_{<t}}$ for all $t \in (1, \text{len}(\mathbf{x}))$ via one forward pass of the original and current model and compute $p^{\text{new}}_{\mathbf{c},\mathbf{x}_{<t}}$ based on the annotation for the given token. For divergences $\mathrm{D}_f$, we consider both the forward and reverse KL divergence, noting that the two encode difference preferences. Minimizing the former is equivalent to minimizing token-level cross entropy, which can be computed analytically. To minimize the latter, we smooth the desired distribution $p^{\text{new}}$ by adding 1e-6 to all elements and renormalizing. We perform gradient-based optimization by summing over the per-sequence losses in a minibatch and calculating the relevant gradients (See Algorithm 1). Note that the Monte Carlo nature of TNT is only to choose which distributions $p_{\mathbf{c},\mathbf{x}_{<t}}$ to update, as the constituent target distributions $p^{\text{new}}_{\mathbf{c},\mathbf{x}_{<t}}$ can be given exactly and thus the divergences computed analytically.

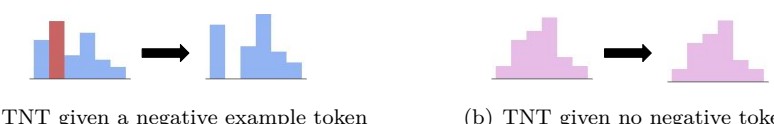

(a) TNT given a negative example token    (b) TNT given no negative token

Figure 1: Summary of Targeted Negative Training (TNT): For negative tokens (i.e. those flagged as undesirable given the preceding tokens), TNT optimizes for a distribution that matches the original, renormalized after the offending token probability is set to zero. For all other tokens, TNT encourages the new distribution to match the original.

## 3.1 The Commutative Property of Negative Updates

A practical benefit of TNT for iterative model updates is that not all negative tokens need to be specified up front. Because of the deterministic nature of the operation to zero out probability mass, one can apply negative examples in any order, both across different $p_{\mathbf{c},\mathbf{x}_{<t}}$ distributions as well as within a given $p_{\mathbf{c},\mathbf{x}_{<t}}$ distribution. This commutative nature of negative updates typically does not apply to "positive" updates—that is, training on samples from the distribution of interest—except for the scenario where the distributions of interest places all their mass on a single token. Negative examples are unique in that they have a known probability mass associated with them under the distribution of interest, i.e. 0. Given an existing distribution and a positive example, on the other hand, there is not enough information to know the probability that an updated distribution should assign to elements in the support of the existing distribution, presumably itself derived from previously training on other positive examples.

---

**Algorithm 1** Targeted Negative Training

1: **Input**: initial model $p^o$ (already trained), inputs $\{\mathbf{c}\}^n_1$, model outputs $\{\mathbf{x}\}^n_1$, token annotations $\{\mathbf{a}\}^n_1$ denoting $\mathbf{x}_t \in \text{supp}(p^{\text{neg}}_{\mathbf{c},\mathbf{x}_{<t}})$
2: $p^m \leftarrow p^o$
3: **for** each iteration **do**
4:    Get $p^m_{\mathbf{c},\mathbf{x}_{<t}}$ for all $\mathbf{c}, \mathbf{x}_{<t}$ in batch (forward pass of $p^m$)
5:    Get $p^o_{\mathbf{c},\mathbf{x}_{<t}}$ for all $\mathbf{c}, \mathbf{x}_{<t}$ in batch (forward pass of $p^o$)
6:    Compute $p^{\text{new}}_{\mathbf{c},\mathbf{x}_{<t}}$ for all $\mathbf{c}, \mathbf{x}_{<t}$ in batch (Equation (2))
7:    Calculate TNT loss (Equation (4))
8:    Calculate gradients for weights in $p^m$ and update $p^m$
9: **end for**
10: **Return** $p^m$

---

## 3.2 Annotating Data for Targeted Negative Training

Ideally, the token-level annotations should align with the autoregressive structure of TNT methods; namely, a negative token should indicate that the subsequence up to and including token $\mathbf{x}_t$ is no longer acceptable such that $p(\mathbf{x}_t|\mathbf{x}_{<t}, \mathbf{c})$ should be 0. For one, this means that not all tokens in an unwanted multi-token

word or expression should necessarily be marked negative—an example includes word or phrase prefixes that could potentially be continued in a manner that is acceptable. For simplicity in the annotations, we make the assumption that for any undesirable phrase, the tokens are undesirable immediately. This is generally a reasonable assumption as long as there are good replacements for the undesirable content which do not overlap in prefix.

In addition, in certain cases a given $p_{\mathbf{c},\mathbf{x}_{<t}}$ will not be relevant in the optimal model. For instance, for a sequence such that $p^{\text{new}}_{\mathbf{c},\mathbf{x}_{<t}}(\mathbf{x}_t) = 0$, the sequence $[\mathbf{c},\mathbf{x}_{\leq t}]$ is out-of-support under $p^{\text{new}}$ meaning $p_{\mathbf{c},\mathbf{x}_{\leq t}}$ is not a relevant distribution. However, in initial study, we found that including such distributions (e.g. all the constituent distributions that occur at time steps after a negative token) still helped constrain the model towards its original. Thus, we chose to include them in the loss.

## 4 Related Work

**Inference-time procedures for controllable generation.** Many works consider alternative decoding strategies to constrain model outputs, either with hard-coded rules such as length penalties and lexical constraints (Paulus et al., 2018; Hokamp & Liu, 2017; Wu et al., 2016; Lu et al., 2021) or auxiliary models such as classifiers or other conditional generative models (Dathathri et al., 2019; Krause et al., 2021; Yang & Klein, 2021; Liu et al., 2021; Meng et al., 2022). These approaches do not change the original model but change the sampling process to effectively sample from an alternative distribution. These approaches can incur non-trivial complexity to the inference process: guided decoding based on rules can result in significant maintenance overhead as the rule set gets more complicated, and controllable generation via auxiliary models involves at least an additional forward pass by the auxiliary model at every decoding time step. In contrast, we propose finetuning approach which does not require a specialized inference pipeline.

**Controllable generation using moment constraints.** The solution of a minimal targeted update is equivalent to that of a pointwise moment constraint defined previously in Khalifa et al. (2020); Korbak et al. (2022a). However, the proposed algorithms differ substantially. Namely, the above works generally consider both distributional and pointwise constraints and employ a two-stage training procedure to build a model to satisfy both: first, they train an energy-based model (EBM) to match the desired solution, and second they train an autoregressive generative model to approximate the distribution implied by the EBM via importance weighting using samples from the model being trained. In contrast, this work considers pointwise constraints only and derives a finetuning procedure which only requires training one model via analytically computable token-level divergences (Section 3). Like previous work, Meng et al. (2022) also consider sequence-level constraints but prove that this setting can be translated into token-level guidance (i.e., relating $p_{new}(x_t|x_{<t},c)$ to $p_0(x_t|x_{<t},c)$) via the approximation of $\Pr_{y\sim p(y|x)}[C(x,y)|y_{<t}]$ for all $y_{<t}$ and sequence-level boolean constraint function $C$. Consequently, Meng et al. (2022) are able to propose a simpler algorithm than existing work, and by combining ideas in their work with this work, it is possible to define a finetuning algorithm that optimizes analytical token-level divergences even given only sequence-level annotations (define $p^{\text{new}}_{\mathbf{c},\mathbf{x}_{<t}}$ using results from Meng et al. (2022), optimize using TNT).

The solution to a minimal targeted update differs from the solution of other objectives which incorporate some form of KL divergence penalty to the loss, as the latter interpolate between the competing objectives of maximizing reward and minimizing KL divergence (Ziegler et al., 2019; Wu et al., 2023; Lu et al., 2022). However, some objectives, e.g., (Ziegler et al., 2019; Wu et al., 2023), can be rewritten as minimizing the reverse KL divergence between the current model and a target distribution that reweights the original model according to $\exp(\frac{1}{\beta}r(\mathbf{x}))$ for sequence-level rewards (see Korbak et al. (2022b); Rafailov et al. (2023)) or $\exp(\frac{1}{\beta}\sum_t r_t(\mathbf{x}_{\leq t}))$ for token-level rewards (see Appendix G), both of which are approximately equal to the solution of minimal targeted update when rewards denote whether a constraint is met and $\beta$ is small.

**Model editing approaches.** Model editing (Cao et al., 2021; Zhu et al., 2020; Hase et al., 2021; Mitchell et al., 2021; 2022) focuses on updating a language model to output a corrected fact (e.g. updating the answer to "Who is the Prime Minister of the UK?" when someone new is appointed), rather than constraining a model to avoid certain generations. In fact, most model editing techniques do not even take into account the negative example (i.e. the outdated or incorrect fact), instead focusing on maximizing the likelihood of correct facts. The reliance on corrected outputs distinguishes the model editing setup from minimal targeted updates.

Whereas corrections may be a natural source of supervision for updating facts, they can be overly prescriptive for tasks such as detoxification where there is a wide range of possible "corrections" to an undesirable output. In addition, maximizing the likelihood over a sample of corrected outputs does not preclude the resulting model from placing non-negligible probability mass over undesirable examples, so a method for avoiding certain outputs can still be useful even when corrections exist.

**Parameter-efficient finetuning.** Parameter-efficient finetuning (Houlsby et al., 2019; Chen et al., 2023; Li & Liang, 2021; Ben Zaken et al., 2022; Hu et al., 2022) is orthogonal to minimal targeted updates, as it is possible to both change a small number of parameters but greatly modify model behavior (as evidenced by the parameter-efficient finetuning literature), as well as change all parameters without changing the model distribution (due to the non-identifiability of neural networks). Parameter-efficient finetuning can be used in tandem with an objective for minimal targeted updates.

## 5 Experiments

We consider two use cases for targeted negative training, reducing hallucinations and toxicity. All experiments utilize T5 base (220M parameters). First, we finetune T5 on the original training set. Then, we generate from the model given training and validation inputs and annotate the generations. Next, we use the annotated generations to update the model. To evaluate, we compute the prevalence of the unwanted behavior among the new model's generations on the test inputs, as well as similarity between the old and new model's generations. We use greedy decoding for all generations.

Next, we describe the datasets and methods. See Appendix B for full dataset and experimental details.

### 5.1 Reducing Hallucinations in Summarization

We train a summarization model using the XSUM dataset (Narayan et al., 2018) consisting of articles as inputs and summaries as outputs. Models trained on this dataset are known to hallucinate in their generations (Ji et al., 2022), and we see the same behavior in the model we train. Using the automated heuristic for detecting hallucination in Nan et al. (2021), we see that approximately 21% of the model's generations contain some form of hallucination. We use the same hallucination detection logic to annotate the model's generations on the training inputs and identify hallucinations in the test generations for evaluation.

### 5.2 Avoiding Toxicity in Response Generation

We train a response generation model using the Civil Comments dataset of online comments (Borkan et al., 2019). Due to the nature of online forums, the comments and responses occasionally contain toxic language. To label text spans as toxic, we train a token-level toxicity classifier on the Civil Comments Spans dataset Pavlopoulos et al. (2021), a subset of the Civil Comments dataset (same splits) where individual spans were labeled "insulting, threatening, an identity-based attack, profane/obscene, or otherwise toxic." We finetune Spacy's CNN-based named entity recognition (NER) model, following Pavlopoulos et al. (2022), and use the finetuned model (65.1/59.5/65.8 F1 on train/val/test) to annotate our language model's generations. Among the initial T5 model's generations, 8.2% contain toxic spans as labeled by our toxicity classifier.

We recognize that our notion of hallucination and toxicity in these tasks is simplistic, as both are based on automated heuristics rather than human annotations and evaluations. However, the goal in these experiments is not to solve the open problems of hallucination and toxicity in text generation but rather to evaluate TNT as a method for producing a minimal targeted update given examples of outputs to avoid, in comparison to other finetuning approaches.

### 5.3 Methods

We consider two baseline finetuning procedures that consider token-level negative signal: negative likelihood for negative tokens and LL for all other tokens in the generations (NL + LL) and unlikelihood for negative tokens and log-likelihood for all other tokens in the generations (UL + LL).[4]

We consider the following targeted negative training methods: Targeted Negative Training Forward-Forward (TNFF) uses the forward KL divergence for both positive and negative signals and Targeted Negative Training Reverse-Reverse (TNRR) uses the reverse KL divergence for both. Targeted Negative Training Reverse-forward (TNRF) uses the reverse KL for negative tokens and forward KL for positive token indices. Finally, to compare more closely to the UL and NL, we consider Targeted Negative Training Forward-LL (TNFLL) and Targeted Negative Training Reverse-LL (TNRLL), which utilize forward and reverse KL divergence for negative tokens and maximum likelihood of the token sample for positive tokens, i.e., a single-sample Monte Carlo estimate of the forward KL divergence.

Following Welleck et al. (2020), we introduce a hyperparameter $\alpha$ on the negative losses for all methods. We consider alpha values 1e-4 to 1e4 (every power of ten). For each method, we perform a hyperparameter sweep for learning rate at $\alpha = 1$ and use the chosen learning rate across all values of $\alpha$. For each run, we perform model selection using validation loss.

### 5.4 Results

**Main results.** Inspired by precision-recall curves, we construct a curve for each objective's similarity score across different rates of unwanted behavior reduction (Figure 2(a) and (c)). Namely, for each method we plot the highest BLEU achieved across $\alpha$ values that achieve less than the given level of hallucination or toxicity. We consider hallucination and toxicity values in increments of 0.1 percentage points. We also plot the composite curve between baseline methods and TNT methods (Figure 2(b) and (d)). Selection is performed on validation data.

For both tasks, we see that the targeted negative training losses allow for a greater flexibility in the trade-off between maintaining similarity with original generations and reducing unwanted behavior. Notably, while the baseline methods cannot achieve a BLEU score above 54 for XSUM and 40 for Civil Comments, regardless of what level alpha is set to, TNT methods which compute an exact divergence on the positive tokens (TNFF, TNRR, TNRF) can trade off how much hallucination or toxicity is reduced to achieve significantly higher BLEU scores. In fact, for the task of reducing toxicity, several of the targeted negative training losses are strictly better than the baseline methods at targeted updates across all levels of toxicity rate reduction. In general, TNFF is the TNT loss that yields the least amount of change overall but can only reduce unwanted behavior up to a certain level, while TNRR and TNRF can yield even further reductions with large enough $\alpha$ while still being more targeted than baselines. Overall, the area under the similarity-reduction curves for TNT methods far outstrips that of the baseline methods (55.5 vs. 44.6 for hallucination, 73.9 vs. 32.9 for toxicity). We include similarity vs. reduction plots for other measures of similarity in Appendix E and see that TNT methods continue to outperform baselines.

Once we also consider the amount of introduced disfluencies (see Appendix D for details and examples), even TNT methods that achieve comparable similarity and reduction rates as baselines are shown to be significantly better at avoiding the introduction of new disfluencies to the generations (see Table 1 for combined similarity and disfluency results at a fixed rate of reduction, and Figure 3 for disfluency results across multiple rates of reduction). Note that TNFLL and TNRLL introduce fewer disfluencies relative to baselines despite only differing their loss terms for negative tokens, empirically corroborating the analysis in Section 2.2 that a targeted negative loss constrains the resulting update in a way existing negative losses do not.

**Increasing model size.** We repeat the hallucination experiment on the 1-billion parameter variant of PaLM-2, a decoder-only model (see Figure 4). For rates of reduction up to 50%, TNT methods offer a better trade-off between similarity and reduction than baseline methods; beyond this point, baseline methods look

---

[4]We find that without LL terms on the non-negative tokens, the model degrades to outputting purely disfluent text after a few steps of finetuning. Our results are corroborated by He & Glass (2020), who also acknowledge that they cannot retain model performance without positive signals.

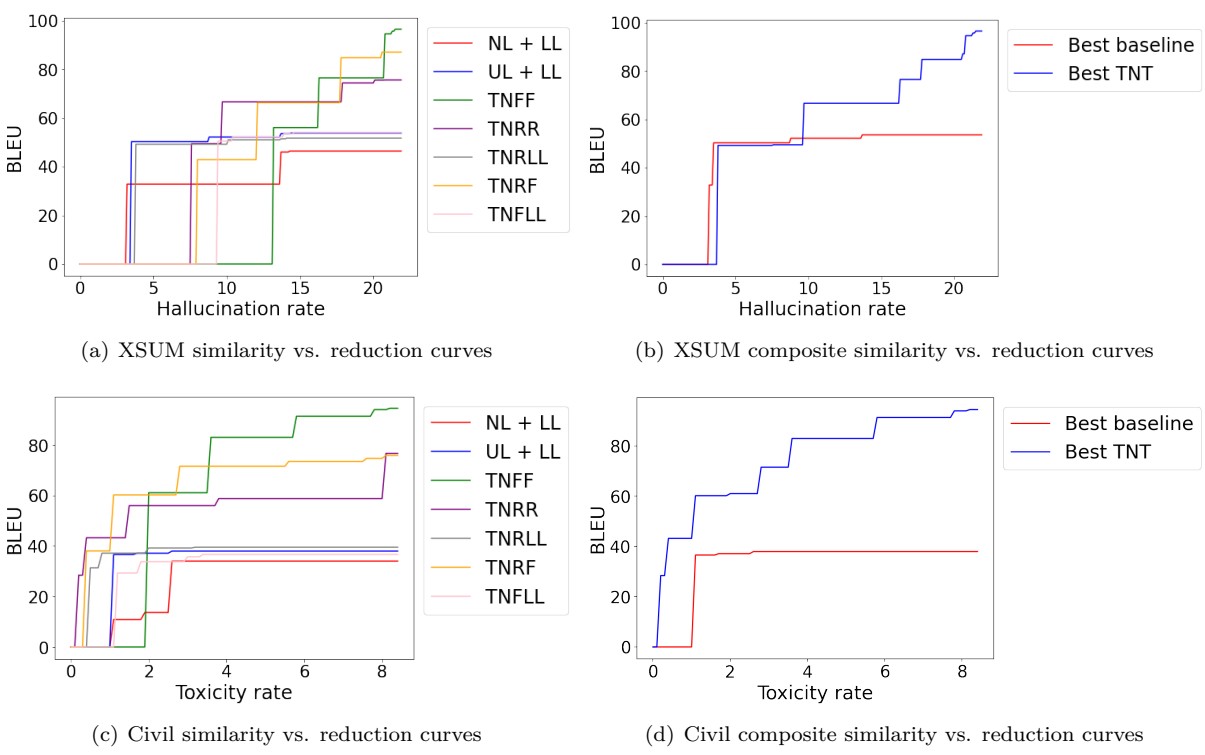

(a) XSUM similarity vs. reduction curves

(b) XSUM composite similarity vs. reduction curves

(c) Civil similarity vs. reduction curves

(d) Civil composite similarity vs. reduction curves

Figure 2: Across nearly all values of reducing unwanted behavior, the suite of TNT objectives is able to achieve a comparable or better trade-off than baseline methods (NL + LL and UL + LL) between reducing unwanted behavior and minimizing change relative to the original model's generations. On the y-axis, higher is better.

better with respect to the similarity vs. reduction trade-off but introduce many more obvious disfluencies. Compared to the results in Figure 2 on T5-base, the results on PaLM 2 are worse for similarity vs. reduction but better for disfluencies vs. reduction. On one hand, the larger and better model seems harder to update. On the other hand, given larger models are generally better to begin with, it arguably becomes even more important to focus on targeting an update versus fully removing unwanted behavior, and this regime is where TNT methods shine over baselines.

**Ablations.** On the toxicity reduction task, we also report the results of an ablation where methods are trained on external data (i.e., labeled spans from the original Civil Comments spans dataset) rather than model generations (Appendix F); all methods yield more targeted updates when using model generations, highlighting the benefit of prioritizing more common token conditional distributions, yet TNT still methods outperform baselines, highlighting the benefit of the proposed losses regardless of the set of token conditionals that are targeted.

We also vary the dataset size used for the update to assess how different TNT methods perform at lower data volumes. Results are presented in Figure 5. Overall, less data results in a smaller reduction in unwanted behavior, and the more a TNT loss constrains the outputs, the better the trade-off with similarity metrics as dataset size decreases. Namely, while TNFLL and TNRLL get strictly worse in both similarity and hallucination as dataset size decreases, TNFF, TNRF, and TNRR are able to better trade-off between similarity and reduction as dataset size decreases. TNFF shows a strictly positive trend implying that generations change less in general, suggesting its promise as a method even at small dataset sizes.

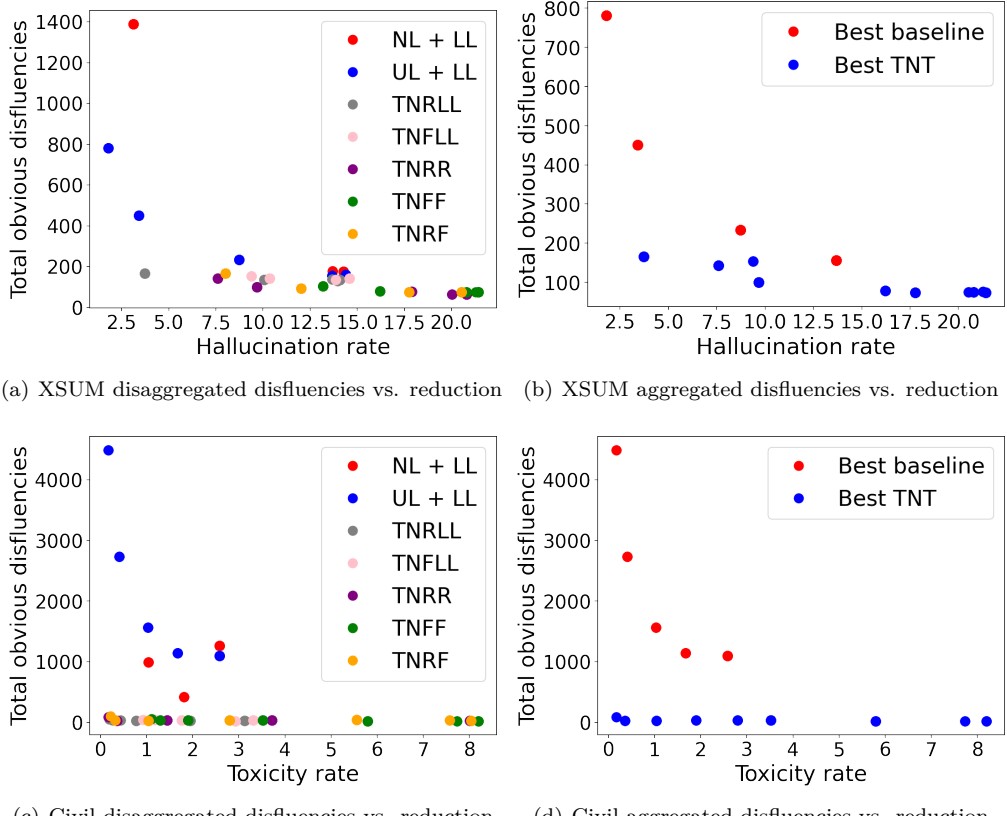

(a) XSUM disaggregated disfluencies vs. reduction     (b) XSUM aggregated disfluencies vs. reduction

(c) Civil disaggregated disfluencies vs. reduction     (d) Civil aggregated disfluencies vs. reduction

Figure 3: Baseline methods introduce more obvious disfluencies (word repeats and random ?? tokens) than TNT methods, especially as the rate of unwanted behavior is reduced to small amounts. For readability and a more direct comparison to Figure 2, only points that are located on the frontier of the similarity vs. reduction curves are plotted.

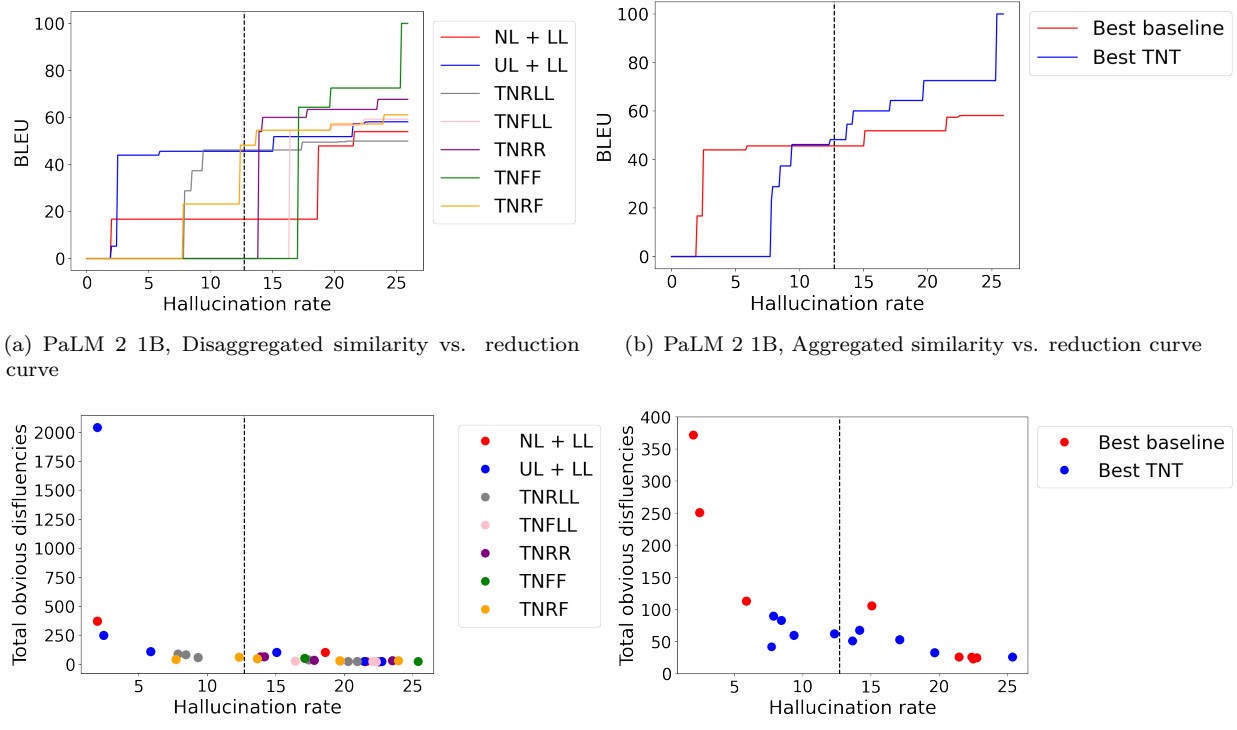

(a) PaLM 2 1B, Disaggregated similarity vs. reduction curve

(b) PaLM 2 1B, Aggregated similarity vs. reduction curve

(c) PaLM 2 1B, Disaggregated disfluency vs. reduction plot

(d) PaLM 2 1B, Aggregated disfluency vs. reduction plot

Figure 4: Results on XSUM and PaLM 2 1B. TNT methods yield a better trade-off between similarity vs. reduction than baselines up to a 50% reduction rate. TNT methods struggle to reduce the hallucination rate past this point, while baseline methods do so but at the expense of increasing obvious disfluencies. For readability and a more direct comparison to Figure 2, only points that are located on the frontier of the similarity vs. reduction curves are plotted.

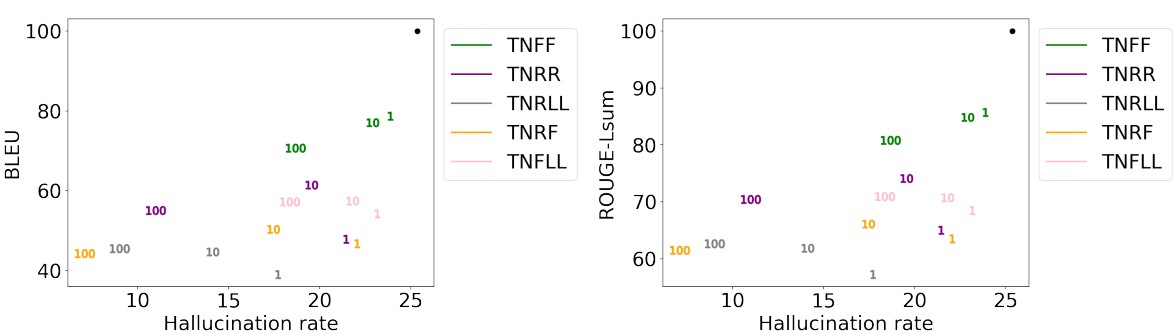

Figure 5: Similarity vs. reduction results as dataset size varies (results are on PaLM 2 1b with $\alpha = 1$). Numbers signify percentage of the original dataset used for training and validation, with test set for evaluation held constant. With less data, TNT methods are generally less effective at reducing hallucination rate. However, TNFF stands out as method that is able to minimize the changes to the original model behavior even with less data (positive slope in the results), suggesting its practical efficacy for minimal targeted updates in low-data regimes. Black dot signifies the original model's metrics.

Table 1: Comparison of methods when the rate of hallucination or toxicity has been reduced by at least 75% (i.e., to less than 5.25% hallucination rate and 2.06% toxicity rate). BLEU, ROUGE-L and Seq Acc measure similarity to the original generations; Hallucination and Toxicity measure the rate of unwanted behavior; and Repeats and Random ?? give counts of obvious disfluencies. Results are bolded if better than all baseline methods. See Appendix D for examples of disfluencies, and Figure 9 for disfluency results for each of Repeats and Random ?? across different reduction rates.

| | BLEU | ROUGE-L | Seq Acc | Hallucination | Repeats | Random ?? |
|---|---|---|---|---|---|---|
| Original | 100.0000 | 100.0000 | 100.0000 | 21.3432 | 76 | 0 |
| NL+LL ($\alpha = 1.0$) | 32.8334 | 52.2349 | 1.5397 | 3.1148 | 1297 | 92 |
| UL+LL ($\alpha = 10.$) | 50.2975 | 65.8047 | 8.2117 | 3.4068 | 127 | 324 |
| TNRLL ($\alpha = 1.0$) | 49.2464 | 64.9200 | 7.3268 | 3.7342 | **99** | **67** |

| | BLEU | ROUGE-L | Seq Acc | Toxicity | Repeats | Random ?? |
|---|---|---|---|---|---|---|
| Original | 100.0000 | 100.0000 | 100.0000 | 8.1830 | 16 | 4 |
| NL+LL ($\alpha = .01$) | 13.6497 | 32.6104 | 2.0661 | 1.8150 | 287 | 136 |
| UL+LL ($\alpha = 1.0$) | 37.1265 | 59.9806 | 20.4405 | 1.6784 | 23 | 1122 |
| TNFLL ($\alpha = 1.0$) | 33.7884 | 57.1009 | 18.1630 | 1.7577 | 36 | **1** |
| TNRLL ($\alpha = 0.1$) | **39.1922** | **61.4776** | **22.9207** | 1.9471 | 23 | **1** |
| TNRR ($\alpha = 0.1$) | **55.9532** | **71.5365** | **35.1366** | **1.4493** | 34 | **3** |
| TNRF ($\alpha = 1.0$) | **60.2071** | **74.8574** | **39.6167** | **1.0396** | **21** | **3** |
| TNFF ($\alpha = 10.$) | **61.0565** | **74.2388** | **40.0749** | 1.9031 | 33 | **3** |

## 6 Discussion

In this work, we propose targeted negative training, a suite of methods for finetuning a language model to avoid unwanted behavior in a targeted fashion, given token-level annotations of the model's generations. While baseline finetuning objectives do not sufficiently constrain how probability mass is dispersed in a negative update, TNT methods directly optimize for a model whose constituent token distributions are the solutions to a minimal targeted update.

Broadly, TNT could be a useful tool for improving the safety of autoregressive generative models by offering a means to iteratively refine a model after it has been initially trained.

TNT is not without its limitations, however. First, TNT requires keeping the original model during training, meaning a larger memory footprint. The additional forward pass through the original model also incurs additional computational cost at each gradient step. Fortunately, however, TNT does not require any extra computational or memory cost during inference. Plus, given the growing interest in finetuning methods that utilize multiple models to regularize towards the original (e.g., RLHF-PPO Ziegler et al. (2019) utilizes three), strategies for mitigating these extra costs during finetuning could be useful broadly.

TNT also requires token-level annotations which may be hard to acquire in certain cases. Next, TNT only targets negative examples that have been specified and could increase the presence of other similar bad words that were present in the generations but not flagged. This result highlights the importance of high-quality annotations. Luckily, the commutative property of negative updates makes it easy to apply TNT iteratively for different sets of negatives to address unwanted behavior as it is noticed. Finally, our experiments show that even though all TNT methods target the same updated model, the choice of objective matters for where on the spectrum between a complete reduction vs. minimal change the resulting model ends up. The methods that allow for the most targeted changes struggle to reach the highest levels of rate reduction and vice versa, suggesting that the optimization strategies in this suite of methods still have room for improvement. For instance, future work could consider other choices of divergences as well as additional optimization tricks to see if it is possible to achieve a more Pareto optimal trade off between reducing unwanted behavior and minimally changing the original model.

## Broader Impact Statement

This work aims to update a language model to reduce the generation of unwanted outputs. However, we note that the experiments consider simplified definitions of unwanted text, and more sophisticated definitions should be annotated and considered in real-world uses of TNT.

## Acknowledgments

This work was partly supported by the NIH/NHLBI Award R01HL148248, NSF Award 1922658 NRT-HDR: FUTURE Foundations, Translation, and Responsibility for Data Science, NSF CAREER Award 2145542, ONR N00014-23-1-2634, and Apple.

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

# A    Optimal Distribution in Targeted Negative Likelihood

Targeted negative likelihood optimizes for the distribution in $p^k \in \mathcal{P}_k$ that minimizes the reverse KL divergence with the original distribution: $p^{\text{new}} = \min_{p^k \in \mathcal{P}_k} \text{KL}(p^k || p^o)$. This distribution is also known as the information projection of the original distribution onto $\mathcal{P}_k$.

To solve for $p^{\text{new}}$, we use Theorem 1 Property A from Khalifa et al. (2020) (also Remark 3.1 in Csiszár & Shields (2004)). We first restate Theorem 1 A from Khalifa et al. (2020) below for readability, and then reiterate the theorem using using notation consistent with the main paper.

To quote Khalifa et al. (2020):

> To recap our formal approach, we have a finite set $X$, a distribution $a$ over $X$ s.t. $a(x) > 0, \forall x \in X$, and real functions $\phi_1, ..., \phi_k$ over $X$. We specify moment constraints $\mu_i = \bar{\mu}_i$ on distributions $c$ over $X$, where $\mu_i \doteq \mathbb{E}_{x \sim c} \phi_i(x)$ and the $\bar{\mu}_i$'s are given targets; the set of distributions satisfying these constraints is denoted by $\mathcal{C}$. Our problem is to find a $p$ such that $p = \arg\min_{c \in \mathcal{C}} D_{\text{KL}}(c, a)$.
>
> ...
>
> **Theorem 1. (A)** *There exists a unique solution $p$ to the problem above, obtained as $p(x) \propto P(x)$ where $P$ is in* exponential family *form:*
>
> $$P(x) = a(x) \; \mathbb{1}[x \in X_{\mathcal{C}}] \; e^{\sum_i \lambda_i \phi_i(x)}. \tag{5}$$
>
> *In other words $p(x) = 1/Z \; P(x)$, with $Z = \sum_{x \in X} P(x)$; $P$ is an unnormalized distribution, i.e. an* EBM. *Here $X_{\mathcal{C}} = \{x \in X | \exists c \in \mathcal{C} \; s.t. \; c(x) > 0\}$ is the "support set" associated with $\mathcal{C}$. The $\lambda_i$'s are real numbers called the* natural parameters *associated with the moments $\mu_i$.*

Reframed in the notation and setting of this work, we have the following: For distribution $a(x)$, we have our original distribution $p_o(x)$. The support set $X_{\mathcal{C}}$ in this setting is the complement of $\text{supp}(p^{\text{neg}})$; thus, for $\mathbb{1}[x \in X_{\mathcal{C}}]$, we have $\mathbb{1}[x \notin \text{supp}(p^{\text{neg}})]$. The moment constraint $\mathbb{E}_{x \sim c}[\phi(x)]$ is a pointwise constraint, namely that $\mathbb{E}_{x \sim p^k(x)}[\mathbf{1}[x \notin \text{supp}(p^{\text{neg}})]] = 1$. Then, since $\phi(x)$ is constant for $x \notin \text{supp}(p^{\text{neg}})$, we have

$$p^{\text{new}}(x) \propto p^o(x)[x \notin \text{supp}(p^{\text{neg}})] \exp(\lambda f(x)) \tag{6}$$
$$\propto p^o(x)[x \notin \text{supp}(p^{\text{neg}})]. \tag{7}$$

In other words, the optimal distribution removes probability at the negative tokens and renormalizes.

# B    Experimental Set Up

## B.1    Dataset Creation.

We use the XSUM dataset (Narayan et al., 2018) for the reducing hallucination task and Civil Comments (Borkan et al., 2019) for the reducing offensive phrases task. We use the datasets themselves for finetuning the base models and generations from the model itself for updating afterward. We describe both in detail below.

**Datasets for Initial Finetuning.** For the hallucination experiment, we use the XSUM train, validation, and test splits. The dataset sizes for train, validation, and test are 203,577, 11,305, and 11,301. For the offensive phrases experiment, we use the Civil Comments dataset of toxic online comments. This dataset is traditionally used for toxicity detection, but here we repurpose the dataset for response generation. In particular, we train our encoder-decoder model to output the text given its parent text, the previous comment the main text is responding to. For the main finetuning dataset, we use only examples that include parent text, decreasing the original dataset size from 1.8 million to 1 million. We use the List of Dirty, Naughty, Obscene, and

Otherwise Bad Words, downloaded from https://github.com/ LDNOOBW/List-of-Dirty-Naughty-Obscene-and-Otherwise-Bad-Words/blob/master/en, as our list of offensive phrases to avoid. In the dataset of 1 million examples, words from the aforementioned list occur in approximately 2% of the targets. To increase the relative proportion of examples with obscenity in the dataset that the model sees to 10%, we subsample the dataset further by randomly removing examples whose outputs do not contain any of the offensive words in the list. The resulting train, validation, and test (unused) sets are of size 175,754, 21,974, and 22,009.

**Filtered Datasets for Finetuning.** To generated the filtered dataset from XSUM (for both finetuning from stratch and from the current model), we use code from Nan et al. (2021) as an automated heuristic to determine whether a hallucination exists in the generated summary. The code uses spacy's named entity recognition to first locate a set of entites from the output, followed by a regex-based matching to determine if the entity is present in the source input.

**Datasets for Finetuning Updates.** After the initial models have been trained (see next section for training details), we generate an output from the model for each input using greedy decoding. We then take the model's outputs and annotate the undesirable tokens using the procedure described in the main paper. For hallucination if an entity detected in the output is not detected in the input, then we marked the entire entity as a hallucination, and if our trained toxicity classifier labels a span in the output as offensive, we mark it as undesirable, even if the input contained the same offensive phrase. We use the train, validation, and test splits for the inputs but the model's own generations for the output. We evaluate all methods on the test set from this process, to compare how much these alternative methods result in deviations from the original model's generations.

## B.2 Training

For all runs, we use a batch size of 32, dropout rate of 0.1, and no label smoothing. For all runs, the cross entropy loss includes the square of the logsumexp of the logits as a penalty, scaled by a factor of 0.0001. For all experiments, we use Google Cloud v4 TPU pods.

For the initial finetuning, we train a base T5 model with learning rate 1e-3 and select the best checkpoint every 10,000 steps based on validation loss. Our resulting models are finetuned for 30,000 steps on XSUM and 40,000 steps on Civil Comments.

For the updates and alternative finetuning, we run a sweep across four different learning rates (1e-3, 1e-4, 1e-5, 1e-6) and choose the best model per every 1,000 steps based on validation loss. We run updates for a total of 100,000 steps for the T5 model, and 200,000 steps for the the PaLM-2 1b model. The learning rates used for the various methods are as follows:

## C   Results from retraining or finetuning on filtered data

Here, we present automated metrics of similarity and hallucination rate on T5-Base and PaLM-2 1b (Table 3 and Table 4 respectively), as well as a sample generation comparison, to highlight that while training on filtered data can reduce the prevalence of unwanted behavior, the resulting model is far from a minimal targeted update of the original model. In contrast, the TNT update methods presented can enable more targeted changes to a model's behavior.

## D   Disfluencies introduced when using existing losses for negative signal

While other forms of disfluencies can exist, we notice two obvious forms in the existing model generations, which we denote word repeats and random ??. We define a word repeat as the repetition of a single word multiple times; note that this definition does not include phrase repeats, so the prevalence of repetition more broadly is likely higher what is reported under 'word repeats.' We define random ?? as the occurrence of any number of question marks preceding and following a space, meaning its presence is not at the end of a sentence but rather in the middle. See Figure 7 and Figure 8 for examples of both word repeats and random ??.

| Dataset | Method | Learning rate |
|---|---|---|
| | Filtered, trained from scratch | 1e-3 |
| | Filtered, trained from current | 1e-4 |
| | UL + LL | 1e-4 |
| | NL + LL | 1e-3 |
| XSUM, T5 Base | TNFF | 1e-3 |
| | TNRR | 1e-3 |
| | TNRF | 1e-4 |
| | TNFLL | 1e-4 |
| | TNRLL | 1e-4 |
| | Filtered, trained from scratch | 1e-4 |
| | Filtered, trained from current | 1e-6 |
| | UL + LL | 1e-5 |
| | NL + LL | 1e-4 |
| XSUM, PaLM-2 1b | TNFF | 1e-4 |
| | TNRR | 1e-4 |
| | TNRF | 1e-4 |
| | TNFLL | 1e-5 |
| | TNRLL | 1e-4 |
| | Filtered, trained from scratch | 1e-4 |
| | Filtered, trained from current | 1e-4 |
| | UL + LL | 1e-3 |
| | NL + LL | 1e-3 |
| Civil Comments, T5 Base | TNFF | 1e-4 |
| | TNRR | 1e-3 |
| | TNRF | 1e-3 |
| | TNFLL | 1e-3 |
| | TNRLL | 1e-4 |

Table 2: Learning rates chosen based on best validation loss from a sweep of learning rates (1e-3, 1e-4, 1e-5, 1e-6). Learning rates for updates were chosen from the runs with $\alpha = 1$ and shared across all $\alpha$ values.

Table 3: A comparison of the original summarization model to ones obtained by retraining or finetuning on data filtered to remove examples with hallucinations. Model is T5-base.

| | BLEU | ROUGE-L | Seq Acc | hallucination rate |
|---|---|---|---|---|
| Original | 100.0000 | 100.0000 | 100.0000 | 21.3432 |
| Filtered & Retrained | 35.9402 | 54.4479 | 2.0087 | 9.3797 |
| Filtered & Finetuned | 47.9839 | 64.0057 | 6.7339 | 13.6271 |

Table 4: A comparison of the original summarization model to ones obtained by retraining or finetuning on data filtered to remove examples with hallucinations. Model is PaLM-2 1b.

| | BLEU | ROUGE-L | Seq Acc | hallucination rate |
|---|---|---|---|---|
| Original | 100.0000 | 100.0000 | 100.0000 | 25.3606 |
| Filtered & Retrained | 44.9260 | 61.2929 | 5.4951 | 14.0077 |
| Filtered & Finetuned | 71.5848 | 80.8366 | 36.0765 | 22.9095 |

| Input | 12 August 2016 Last updated at 11:51 BST

The report was commissioned by former PM David Cameron in the wake of the Tower Hamlets election fraud scandal in 2015.

He said the review had heard evidence that in a London borough at the general election someone arrived at the polling station with 500 postal votes.

"I'm very worried - I don't believe political parties should touch or handle in any way a completed ballot paper," he said. |
|---|---|
| Original | A review into the way postal votes are handled in the UK has found that the electoral system is not safe. |
| Retrained | A review into electoral fraud has found that voters were handed ballot papers with a "significant" number of postal votes. |
| Finetuned | A review into electoral fraud has found that a voter may have contacted the polling station with a ballot paper. |

Figure 6: Comparison of an example generation from the original model, a model retrained on filtered data, and the original model finetuned on filtered data. Model is T5-base.

| Input | The summertime routes were due to end in late September but due to poor seat sales, the airline is stopping the service at the end of August.

The airline said it had reallocated its planes to "routes with greater demand".

It said it would offer a "full refund or alternative flights" to customers who were booked to fly in September

"We apologise sincerely for the inconvenience caused," said an Aer Lingus statement. |
|---|---|
| Original | Aer Lingus has cancelled all flights between the UK and the Republic of Ireland due to a lack of passengers. |
| NL + LL | Aer Lingus is to stop flying to the the the the the the the the the the the country is based on. |
| TNFLL | Aer Lingus has cancelled all flights between the two countries due to a lack of capacity. |
| TNRLL | Aer Lingus has cancelled all of its summer flights from the country to the mainland. |

Figure 7: Example of a word repeat. The original offending text, as well as the word repeat disfluency, has been underlined.

| Input | "Thanks for pointing out my error, anyhow." _oldgit_gittinolder

Not a problem and yes, I knew what you meant - just couldn't pass up a feeble attempt at humor |
|---|---|
| Original | I'm sorry, but I'm not a troll. |
| UL + LL | I'm sorry, but I'm not a ?? inbot. |
| TNFLL | I'm sorry, but I'm not a joker. |
| TNRLL | I'm sorry, but I'm not a 'oldgit'. |

Figure 8: Example of a random ??. The original offending text, as well as the random ?? disfluency, has been underlined.

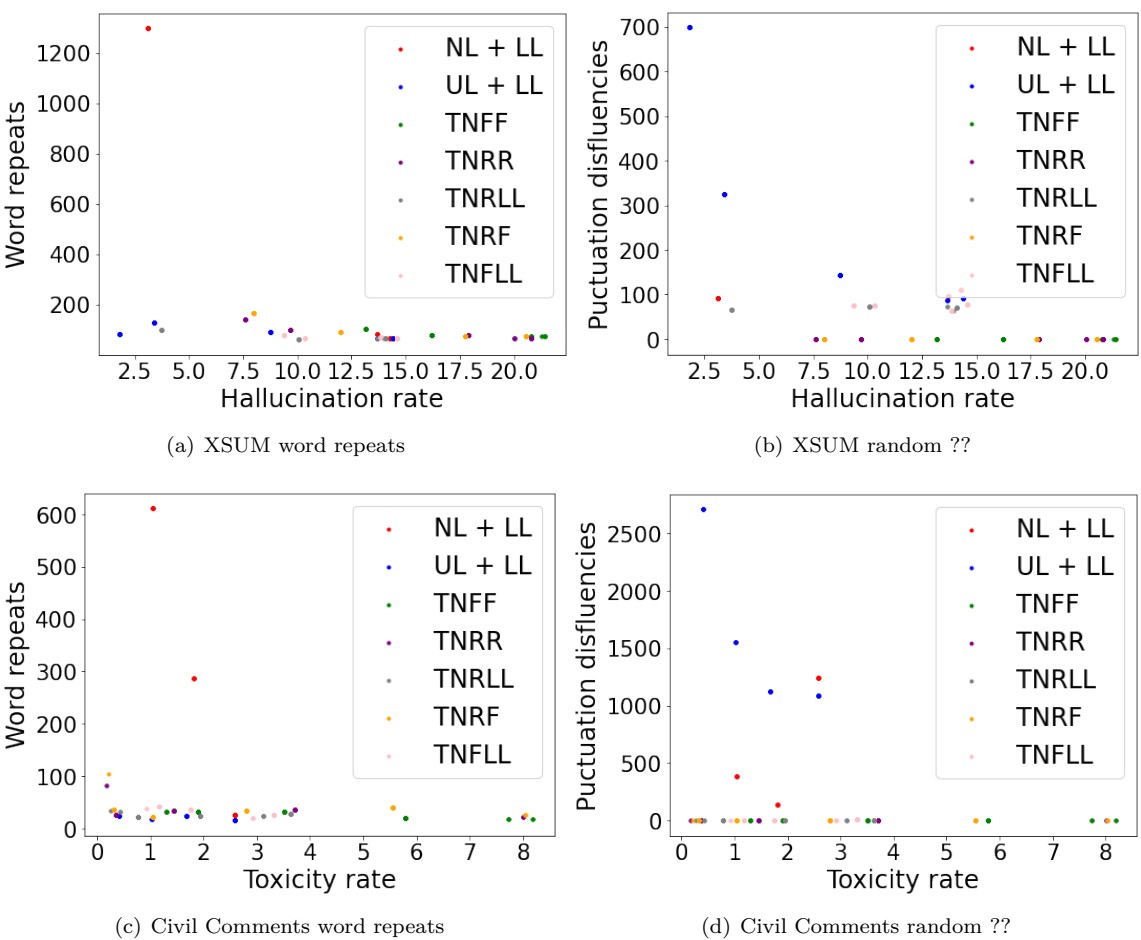

(a) XSUM word repeats

(b) XSUM random ??

(c) Civil Comments word repeats

(d) Civil Comments random ??

Figure 9: Number of disfluencies for different rates of unwanted behavior reduction. For each method, only models that are used in the BLEU similarity vs. reduction curves are plotted here (i.e., the models that are best at maintaining similarity with the original for a given rate of reduction). This choice makes it easy to directly analyze this plot in conjunction to the main figure (Figure 2). From these plots, it is easy to see that baseline methods introduce disfluencies at high rates as they reduce unwanted behavior, where NL + LL tends to introduce repetition while UL + LL tends to introduce random ?? disfluencies. In contrast, the number of disfluencies introduced by TNT methods in both categories combined is orders of magnitudes smaller than the total number introduced by baseline methods.

# E   Main Results with alternative similarity metrics

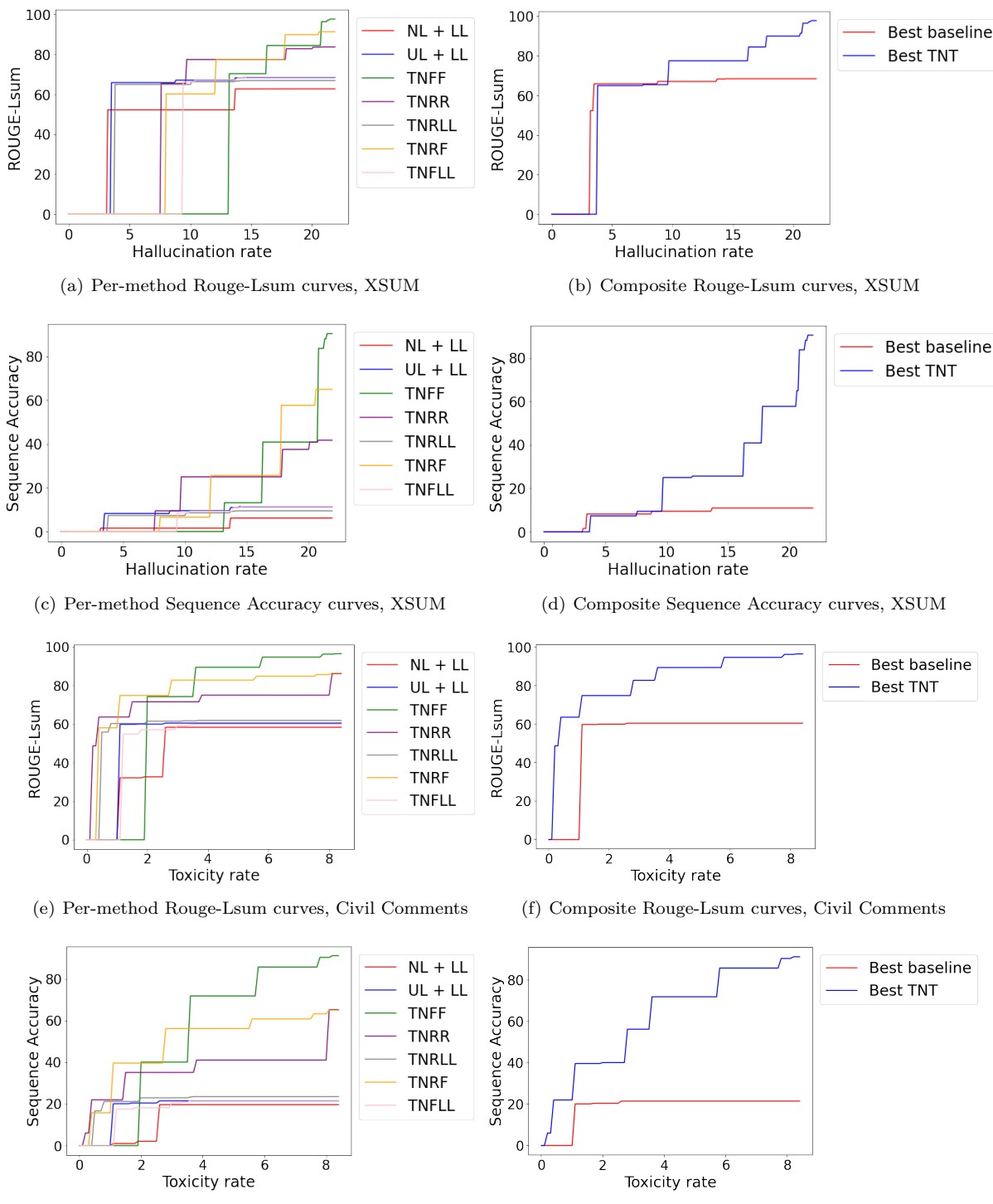

(a) Per-method Rouge-Lsum curves, XSUM

(b) Composite Rouge-Lsum curves, XSUM

(c) Per-method Sequence Accuracy curves, XSUM

(d) Composite Sequence Accuracy curves, XSUM

(e) Per-method Rouge-Lsum curves, Civil Comments

(f) Composite Rouge-Lsum curves, Civil Comments

(g) Per-method Sequence Accuracy curves, Civil Comments

(h) Composite Sequence Accuracy curves, Civil Comments

Figure 10: Alternative similarity measures to BLEU (ROUGE, Sequence Accuracy) show the same trend as the main result in Figure 2: TNT methods yield a better trade-off than baseline methods between reducing unwanted behavior and maintaining similar generations to the original model.

## F  Training with model generations vs. external data

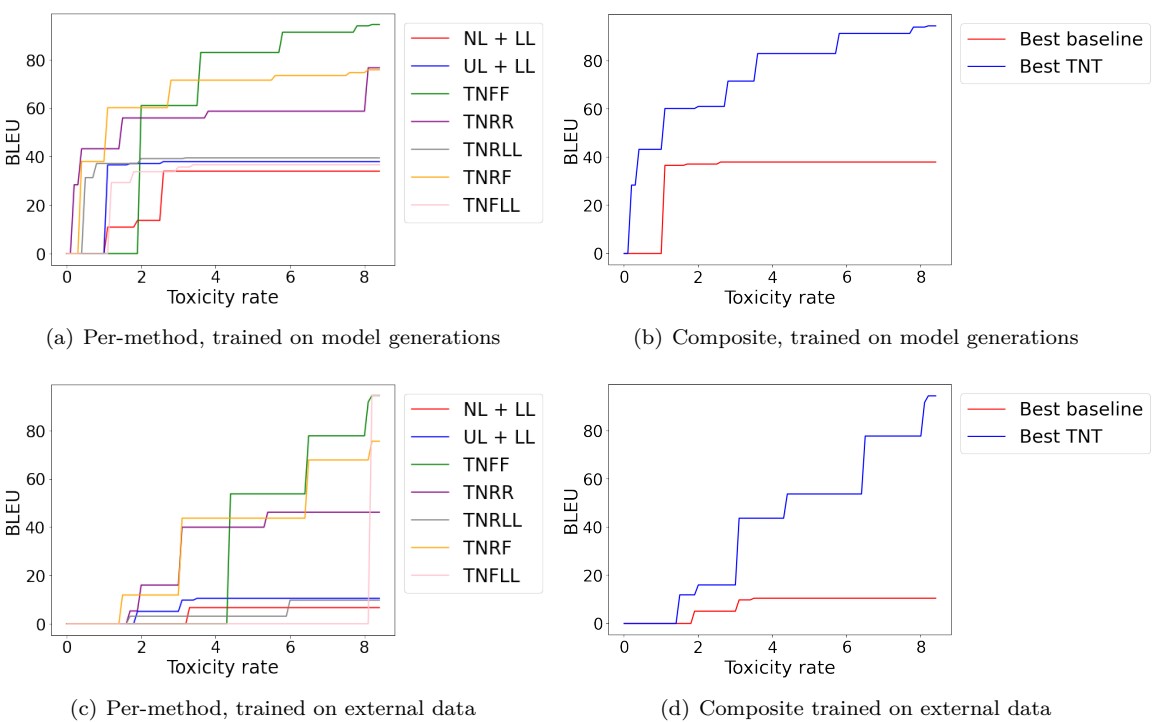

(a) Per-method, trained on model generations

(b) Composite, trained on model generations

(c) Per-method, trained on external data

(d) Composite trained on external data

Figure 11: Similarity vs. toxicity rate reduction curves for the Civil Comments response generation task, comparing a model finetuned on its own generations (top, AUC is 73.9 for TNT vs. 32.9 for baselines) vs. external data (bottom, AUC is 41.9 for TNT vs. 7.4 for baselines). The model stays closer to its original using its own generations for finetuning rather than the data it was trained on.

## G  Token-level KL-constrained RL as a divergence minimization

Here, we show that a token-level version of the objective in Wu et al. (2023) is equivalent to minimizing reverse KL divergence between the policy model $p_\theta(\mathbf{x}|\mathbf{c})$ and a target distribution $p_{\text{new}}(\mathbf{x}|\mathbf{c}) \propto p_o(\mathbf{x}|\mathbf{c}) \exp(\frac{1}{\beta} \sum_t r_t(\mathbf{x}_{\leq t}))$:

$$\max_\theta \mathbb{E}_{p(\mathbf{c})} \mathbb{E}_{\mathbf{x} \sim p_\theta(\mathbf{x}|\mathbf{c})} \Big[ \sum_t r_t(\mathbf{x}_{\leq t}) - \beta[\log p_\theta(x_t|x_{<t}, \mathbf{c}) - \log p_o(x_t|x_{<t}, \mathbf{c})] \Big] \tag{8}$$

$$= \max_\theta \mathbb{E}_{p(\mathbf{c})} \mathbb{E}_{\mathbf{x} \sim p_\theta(\mathbf{x}|\mathbf{c})} \Big[ \sum_t \log[\exp(\frac{1}{\beta} r_t(\mathbf{x}_{\leq t}))] - \log p_\theta(x_t|x_{<t}, \mathbf{c}) + \log p_o(x_t|x_{<t}, \mathbf{c}) \Big] \tag{9}$$

$$= \min_\theta \mathbb{E}_{p(\mathbf{c})} \mathbb{E}_{\mathbf{x} \sim p_\theta(\mathbf{x}|\mathbf{c})} \Big[ \sum_t \log \frac{p_\theta(x_t|x_{<t}, \mathbf{c})}{p_o(x_t|x_{<t}, \mathbf{c}) \exp(\frac{1}{\beta} r_t(\mathbf{x}_{\leq t}))} \Big] \tag{10}$$

$$= \min_\theta \mathbb{E}_{p(\mathbf{c})} \mathbb{E}_{\mathbf{x} \sim p_\theta(\mathbf{x}|\mathbf{c})} \Big[ \log \prod_t \frac{p_\theta(x_t|x_{<t}, \mathbf{c})}{p_o(x_t|x_{<t}, \mathbf{c}) \exp(\frac{1}{\beta} r_t(\mathbf{x}_{\leq t}))} \Big] \tag{11}$$

$$= \min_\theta \mathbb{E}_{p(\mathbf{c})} \mathbb{E}_{\mathbf{x} \sim p_\theta(\mathbf{x}|\mathbf{c})} \Big[ \log \frac{p_\theta(\mathbf{x}|\mathbf{c})}{p_o(\mathbf{x}|\mathbf{c}) \exp(\frac{1}{\beta} \sum_t r_t(\mathbf{x}_{\leq t}))} \Big] \tag{12}$$

$$= \min_\theta \mathbb{E}_{p(\mathbf{c})} \text{KL}\Big( p_\theta(\mathbf{x}|\mathbf{c}) || p_{\text{new}}(\mathbf{x}|\mathbf{c}) \Big). \tag{13}$$

