# OpenReview forum: "Towards Minimal Targeted Updates of Language Models with Targeted Negative Training"
_TMLR — Accepted by TMLR_

### Review · Reviewer_Gka3 · 2024-02-25

**Summary Of Contributions:**

This work proposes a method to update a trained generative model in a targeted way to stop if from outputting unwanted behaviours, while not allowing the rest of the output distribution to change too much. Such a method has obvious and important applications; unwanted behaviour of LLMs for example show up constantly, and re-training or fine-tuning a model to mitigate this can have negative impact on the generation quality plus is expensive. I expect the method proposed in this work to be very valuable to the community and hence this work is a timely contribution.

The method is motivated in great depth, and the proposed targeted update method tackles most of the issues with prior methods that aim to do the same. The authors experiment in the domains of reducing hallucinations in summarisation and reducing toxicity in response generation. They show for a encoder-decoder model of 220M parameters (T5) that their method outperforms baselines (unlikelihood fine-tuning) for most levels of reductions of unwanted behaviour, by quite a large margin (much higher BLEU/ROUGE for high levels of reduction).

**Audience:**

Yes

**Broader Impact Concerns:**

N.A.

**Claims And Evidence:**

Yes

**Requested Changes:**

- Would strengthen the work: experiments with one more model of a different size (e.g. 1B parameters)
- Would strengthen the work: experiments with a decoder-only model
- Would strengthen the work: human evaluation of the method compared to a baseline
- Would strengthen the work: discussion about the increased computational complexity of this method over baselines

**Strengths And Weaknesses:**

**Strengths**
- The proposed method is simple, widely applicable, and a timely and important contribution.
- The motivation and explanation of problems with prior work attempting to do the same is excellent, a pleasure to read and informative.
- The description of the method is well-written and properly contextualised w.r.t. prior work.
- The authors experiment with multiple domains, compare to two baselines, and show that their method significantly improves over the baselines for most levels of unwanted behaviour reduction for two types of annotated data (model-generated and existing).

**Weaknesses**
- For a proposal of such a generally applicable method, the experiments section of the paper seems a bit of an afterthought. The experiments don't come very close to the generative models that are most used currently; decoder-only models with many more parameters. I understand that fine-tuning models with more than 50B parameters is unfeasible, but at least fine-tuning models with two different sizes to see how the method scales would make sense to me. Additionally, it would be interesting to see the same experiments done for a decoder-only model.
- Given the well-documented downsides of automated evaluation metrics like BLEU and ROUGE (and basically any ground-truth overlap metric), it would be insightful to see results of a (small) human evaluation study. Particularly since one of the motivations of this work is that prior methods cause outputs that correlate with unwanted behaviour to be affected as well, and that this method should mitigate that. ROUGE and BLEU doesn't highlight whether such a goal is achieved, but human evaluation could. That said, the BLEU/ROUGE increase w.r.t. the baselines is quite large for large amounts of reduced unwanted behaviour, so I'm not too worried about this, it would just be interesting.
- Some discussion about the increased cost of this method over unlikelihood training would be insightful, since it does introduce an extra forward pass which could be non-neglible for larger models.

---

> ### Author Response · Authors · 2024-04-06
> **Thank you for your review!**
>
> Thank you for your review! We really appreciate your feedback and made the following changes, described below:
>
> **I understand that fine-tuning models with more than 50B parameters is unfeasible, but at least fine-tuning models with two different sizes to see how the method scales would make sense to me. Additionally, it would be interesting to see the same experiments done for a decoder-only model.**
>
> Agreed! Resource constraints unfortunately prevent us from operating at extremely large scales, but we were able to add an experiment with a 1 billion parameter decoder-only model (PaLM-2 1b) that mirrors the existing hallucination experiment. A link to the results can be found [here](https://imgur.com/a/ykMtnLl). We ran a sweep over the same alpha parameter values, comparing (7 methods) $\times$ (9 values of $\alpha$) = 63 finetuning updates runs at the 1 billion parameter size. Given a starting point where 25.36% of generations contain a hallucination, TNT methods yield a better trade-off between similarity vs. reduction than baselines up to a 50% reduction rate. TNT methods struggle to reduce the hallucination rate past this point, while baseline methods do so but at the expense of substantially increasing obvious disfluencies (namely, repetitions). On one hand, the ability to flexibly update the model across different rates of hallucination seems harder with the larger decoder-only model, potentially due to the model being more peaked and better fit to begin with (e.g., cross entropy loss on original data is magnitudes lower). On the other hand, given larger models are generally better to begin with, it arguably becomes even more important to focus on targeting an update versus fully removing unwanted behavior, and this regime is where TNT methods shine over baselines.
>
> **It would be insightful to see results of a (small) human evaluation study. Particularly since one of the motivations of this work is that prior methods cause outputs that correlate with unwanted behaviour to be affected as well, and that this method should mitigate that... That said, the BLEU/ROUGE increase w.r.t. the baselines is quite large for large amounts of reduced unwanted behaviour, so I'm not too worried about this, it would just be interesting.**
>
> We agree that this would be interesting to look into. Resource constraints unfortunately prevent us from running a human evaluation (given hallucinations are particularly expensive to evaluate for wide-ranging topics and the need for IRB approval for toxicity experiments). That said, to your point of “showing that prior methods cause outputs that correlate with unwanted behaviour,” Figure 3 in the updated draft shows that baselines introduce obvious disfluencies like word repetition or substitution with a nonsensical token like `??` much more often than TNT methods do (see [here](https://imgur.com/a/7NBzADb) for easy viewing). Plus, in our new experiments with PaLM 2, we see that even though the overall number of disfluencies has decreased across methods, baselines continue to introduce more disfluencies than TNT methods.
>
> **Some discussion about the increased cost of this method over unlikelihood training would be insightful, since it does introduce an extra forward pass which could be non-neglible for larger models.**
>
> Great point. We have added the following notes to the discussion next to the existing point that TNT has a larger memory footprint during training:
> - (As mentioned by the reviewer) TNT requires an additional forward pass of the original model, increasing the computational cost during training.
> - Given the growing interest in finetuning methods that utilize multiple models to regularize towards the original (e.g., RLHF-PPO typically utilizes three), strategies for mitigating these extra costs could be useful broadly.
> - Fortunately, TNT does not require any extra computational or memory cost during inference.
>
> Once again, thank you for your review! We really appreciate all the great feedback you provided and hope the reviewer finds the additions we have made to satisfy your primary concerns.

---

### Review · Reviewer_yKYh · 2024-03-09

**Summary Of Contributions:**

This paper studies the method to update the LM to avoid unwanted outputs while minimally changing model behavior otherwise. The authors propose a new fine-tuning strategy called Targeted Negative Training (TNT) which minimizes the divergence between the model distribution and desired distribution for every token. The idea of annotating the negative tokens w.r.t. a certain range of context is very interesting and more desirable compared to the previous approaches that treat all tokens in the sentence as unwanted.

**Audience:**

Yes

**Broader Impact Concerns:**

I think the way the authors define toxicity may need a statement.

**Claims And Evidence:**

Yes

**Requested Changes:**

1. Is "random ??" a compilation error?
2. The results compares the best TNT performance with the best baseline performance and demonstrate the proposed method is better than the baseline. However, we don't know how to choose the optimal method in practice, right? I wonder if the author could do some separate plot and explain which method people should choose in practice, especially in the case where the baseline curve outperforms the TNT single method curve at small toxicity/hallucination rate.
3. Could the author explain how they annotated the data with unwanted spans? Is it an objective or subjective process? Any metrics?
4. I don't quite understand the x axis in the figures. For example, in Figure 2, are the rate of percentages? Why the toxicity rates of (a), (b), are of the same scale but the rates of (c) and (d) are different? (It's 0-8 for (c) but (0-0.08) for (d), which I guess is a typo.)
5. Assuming the Figure 2 shows hallucination rates in percentages, I wonder if TNT only outperforms the baselines at larger hallucination rates. If so, Table 1 only shows comparison at a hallucination or toxicity reduction rate of 75% is beyond the limit of the Figure 2 that is supposed to have great benefits. I wonder if the author could provide table numbers for smaller rates (which is more desirable in practice), e.g., at 5%, 10%, and 15% hallucination rates and 1%, 2%, and 4% toxicity rates.

**Strengths And Weaknesses:**

**Strengths**
1. The paper studies an important challenge that exists in alignment fine-tuning: we want to align the model to avoid outputting unwanted phrases but we want to maintain the model's original behavior on normal text.
2. The proposed method poses a more fine-grained control on mitigating the negative outputs while keeping the model otherwise the same.
3. The paper is well-written and easy to follow. It discussed the reasons why the previous approaches do not work well on addressing the challenge and then introduced the new method that can better address the challenge.

**Weaknesses**
1. The results are not very comprehensive and the settings need to be further clarified. See "requested changes" section.

---

> ### Author Response · Authors · 2024-04-06
> **Thank you for your review!**
>
> Thank you for your feedback! Addressing each comment:
>
> **Is "random ??" a compilation error?**
>
> By "random ??," we mean that the resulting model outputs the token `??` in the middle of the sentence, an example of an undesirable change resulting from the negative training particularly of baseline methods. See [here](https://imgur.com/a/8n6xt3I) (also in the appendix) for an example.
>
> **I wonder if the author could do some separate plot and explain which method people should choose in practice, especially in the case where the baseline curve outperforms the TNT single method curve at small toxicity/hallucination rate.**
>
> Thanks for the great suggestion! We added new plots to the main paper which combined with the existing Figure 2 should help guide people to understand what to use in practice. First, Figures 2, [3](https://imgur.com/a/7NBzADb), and [4](https://imgur.com/a/ykMtnLl) illustrate why one generally prefers a TNT method over a baseline method even at small toxicity/hallucination rates; baseline methods introduce many more obvious disfluencies especially at small toxicity/hallucination rates.
>
> As for which method to choose in practice, the disaggregated plots in the Figures 2 and 3 show which TNT methods perform best at different levels of reduction. In general, TNFF is able to best constrain the updated model towards the original, though at the cost of being able to reduce unwanted behavior the least. For settings where a targeted change is more important than completely removing unwanted behavior, this method is probably the best choice. If further reduction of unwanted behavior is desired, then switching to reverse KL divergence on the negative tokens, either with TNRR or TNRF, is recommended.
>
> It will generally be application specific whether preserving initial model behavior is more important or avoiding unwanted outputs is more important, but as a rule of thumb, the former is probably more important for applications where existing models are already useful and usable but can be improved to avoid certain unwanted behavior, while the latter is more important for applications where a model is not yet usable unless it is improved.
>
> **Could the author explain how they annotated the data with unwanted spans? Is it an objective or subjective process? Any metrics?**
>
> The hallucination spans were based on a NER-model and regex-based heuristic from Nan et al 2021, while the toxicity spans were labeled from a token-level toxicity classifier trained on the Civil Comments spans dataset, following Pavlopoulos et al. 2022. The annotation process (and corresponding evaluation process) is automated and thus imperfect (the toxicity classifier F1 is 65.81 on held-out data; we unfortunately do not have metrics for the hallucination heuristic of Nan et al 2021, as computing them requires ground truth hallucination annotations). Fortunately, the limitations of these processes are orthogonal to the goal of the paper, which is to evaluate methods for finetuning on negative annotations in general, regardless of the definition of negative. Thank you for the question; we have added additional details, including this explicit caveat, to sections 5.1 and 5.2.
>
> To address your concern that “the way the authors define toxicity may need a statement,” we defer to Pavlopoulos et al. (2022), the creators of the Civil Comments span dataset, where toxic spans are defined as those that the majority of rates found to be “insulting, threatening, identity-based attack, profane/obscene, or otherwise toxic.”
>
> **Figure 2, rates or percentages? Typo?**
>
> The x-axes are percentages, except for Figure 2d where there is indeed typo. Thank you for catching that typo! Fig 2d has been corrected.
>
> **I wonder if TNT only outperforms the baselines at larger hallucination rates... Table 1 only shows comparison at a hallucination or toxicity reduction rate of 75% is beyond the limit of Figure 2... I wonder if the author could provide table numbers for smaller rates (which is more desirable in practice), e.g., at 5%, 10%, and 15% hallucination rates and 1%, 2%, and 4% toxicity rates.**
>
> Thank you for the question. To clarify, Table 1 shows results at a reduction rate of 75%, which means at most a 5.25% hallucination rate (i.e., 25% of a 21% starting rate) or 2.05% toxicity rate (i.e., 25% of an 8.2% starting rate). Apologies for any confusion; we have updated the caption to the following to make this point clearer: “Table 1: Comparison of methods when the rate of hallucination or toxicity has been reduced by at least 75\% (i.e., to less than 5.25\% hallucination rate and 2.06\% toxicity rate).”
> Instead of producing a different table for each reduction rate, we added [Figure 3](https://imgur.com/a/7NBzADb) which shows the prevalence of disfluencies across different rates. Figure 6 shows that at small hallucination and toxicity rates, baseline methods introduce substantially more disfluencies than TNT methods do.
>
> Thank you again for your review!

---

### Review · Reviewer_tscQ · 2024-03-23

**Summary Of Contributions:**

The paper discussed related methods, particularly on NL+LL and UL+LL which do not constrain the target distribution close to the original distribution of a language model. The paper then proposes a finetuning method namely Target Negative Training (TNT) for pushing down the probability of undesired generation and maintaining the probability of the other generation similar to the original model using KL divergence. Meanwhile, the paper also proposed the data annotation for the pretraining. Finally, the paper shows the proposed method TNT outperforms NL+LL and UL+LL.

**Audience:**

Yes

**Claims And Evidence:**

Yes

**Requested Changes:**

As weakness.

For weakness 1, I regard minimizing unwanted behavior as a kind of human preference. These days there are many works related to reinforcement learning for human preference alignment. I think they can be potentially considered as baselines. On the other hand, the related work in this paper discussed related methods. Why not use them as baselines? I suggest the author at least add reasons for not considering them as baselines.

For weakness 2, the delivery of sample complexity, e.g., how many samples are needed to make the proposed method work, can benefit the following works and the community.

**Strengths And Weaknesses:**

Strength:

1. The method is straightforward and makes sense to me. By adding loss for constraining the output distribution close to the target distribution, the model should reduce the probability of generating negative samples while maintaining the original generative distribution as much as possible.

2. The writing is generally easy to follow with descriptions of implemental details such as that Section 3 mentions adding $1e^{-6}$ for target distribution, and Section 3.2 mentions empirically the author found including those constituent distributions after a negative token helps constrain the model toward original.

Weakness:

1. While many related works are discussed in Section 4, why they are not considered baselines?

2. Section 3.2 mentions including those constituent distributions after a negative token helps. From this point, I guess the performance can be boosted with more samples. Because it seems even those constituent distributions after a negative token can help. Thus, a sample complexity analysis can be added to ablation study.

---

> ### Author Response · Authors · 2024-04-06
> **Thank you for your review**
>
> Thank you for your review! Addressing your comments below:
> ## On related work:
>
> **These days there are many works related to reinforcement learning for human preference alignment. I think they can be potentially considered as baselines.**
>
> Thank you for the comment. While finetuning with preference data is a topic of increased interest in the community, preference data provides less direct information about what is unwanted than negative labels; namely, knowing that a sequence is dispreferred over another sequence does not tell you whether the dispreferred sequence should be slightly less likely or avoided altogether. Negative examples give this information directly, which can be especially useful for settings such as increasing factuality. A method such as TNT that takes into account negative examples can then target changes (i.e., avoiding outputs) that would be much more difficult for a preference alignment method via preference data. For instance, even the optimal solution to objectives such as KL divergence-constrained RL finetuning and Direct Preference Optimization, i.e., $p^*(y|x) \propto p_o(y|x)\exp(r(x, y))$, will not “zero out” (including in a soft manner) a negative example that is likely under the original model unless its reward is extremely negative relative to other examples in the model’s support.
>
> **The related work in this paper discussed related methods. Why not use them as baselines?**
>
> Thank you for the question. The reason we do not compare against the methods mentioned in the related work is because they operate in a different setting from TNT based on either the type of method or data considered. Concretely, going in order of the subheadings in the related work:
> 1. Inference-time procedures change the decoding for a fixed model, whereas TNT is a finetuning method, which confers practical benefits such as constant inference complexity under iterative updates.
> 2. Moment constraint methods utilize sequence-level annotations, whereas TNT utilizes token-level annotations. Where the annotation effort is similar (e.g., labeling “has hallucination” requires identifying hallucinations), an approach like TNT can take advantage of more fine-grained feedback. Otherwise, these methods can be utilized additively (i.e., TNT for token-level annotations, moment constraint methods for sequence-level annotations).
> 3. Model editing techniques rely on correction data, whereas TNT operates on negative examples without the need for corrections, which can be expensive to collect.
>
> In summary, we focus the experiments on methods that fall under the same setup as TNT (i.e., finetuning on token-level labels of negative examples), whereas the methods in the related work span other setups (e.g., decoding vs. finetuning, sequence-level vs. token-level, corrections vs. negative examples). While we attempt to focus this paper on validating the proposed suite of methods in its specific setting, we agree with the reviewer’s sentiment that it could be useful to compare across setups in future application-specific efforts where comparisons between inference and finetuning efforts and different forms of annotations can be more precisely defined.
>
> ## On sample complexity:
> **Section 3.2 mentions including those constituent distributions after a negative token helps. From this point, I guess the performance can be boosted with more samples. Because it seems even those constituent distributions after a negative token can help. Thus, a sample complexity analysis can be added to ablation study… The delivery of sample complexity, e.g., how many samples are needed to make the proposed method work, can benefit the following works and the community.**
>
> Thank you for the suggestion! We have added to the appendix an ablation based on training data size (1%, 10%, 100% of the XSUM dataset containing ~200k training examples). See [here](https://imgur.com/a/SMzyHef). Summarizing the results briefly, with less data all methods are less effective at reducing unwanted behavior. However, while most TNT methods generally see downward curving slopes as dataset size decreases (i.e., less reduction but also more overall change), the trend for TNFF is up and to the right, signifying less reduction is associated with less change overall. In other words, TNFF is a good choice for a minimal targeted update in the low data regime.
>
> Thank you again for your review!

---

### Author Response · Authors · 2024-04-06
**Overall response**

Thank you to all the reviewers; we really appreciate the time you spent sharing your feedback. To highlight the work’s strengths identified by the reviewers:
1. Gka3 states that the method “has obvious and important applications” and is “a timely contribution.” In addition, “the motivation and explanation of problems with prior work …is excellent, a pleasure to read and informative,” and “the method significantly improves over the baselines for most levels of unwanted behaviour reduction for two types of annotated data.”
2. yKYh says that the work “studies an important challenge” and “poses a more fine-grained control on mitigating the negative outputs”, and additionally compliments the paper for being “well-written.”
3. tscQ notes that the proposed method is “straightforward” and “out-performs” while the writing is “easy to follow with descriptions of implementation [sic] details.”

We agreed with all of the reviewers’ requested changes and addressed them by running over 100 additional finetuning jobs, adding three additional figures to the main paper, among other changes to the paper. Most notably:
1. In response to Gka3’s request for experiments on a larger model and decoder-only architecture, we added results on a 1B decoder-only model PaLM-2 (66 additional finetuning experiments, plus initial finetuning and hyperparameter sweeps). See figure [here](https://imgur.com/a/ykMtnLl).
2. In response to yKYh’s request to show more results akin to Table 1 at smaller rates of hallucination/toxicity, we added a figure to the main results that plots the number of disfluencies across rates of unwanted behavior. See figure [here](https://imgur.com/a/7NBzADb).
3. In response to tscQ’s request to analyze the sample complexity of the method, we added an ablation with varying sample sizes. See figure [here](https://imgur.com/a/SMzyHef)).

We detail other specific updates and changes as responses to the individual reviews. We hope the reviewers appreciate our updates and response and are willing to provide a positive recommendation for this work. Thank you!

---

### Decision · Action_Editor_DEC9 · 2024-05-14

**Recommendation:** Accept as is

**Comment:**

This paper proposes Targeted Negative Training (TNT), a method for updating language models to avoid unwanted outputs while minimally changing the model's behavior. (Similar to the recent RLHF setting, but only with negative samples.) Most of the reviewers found the method well-motivated and empirically validated, recommending acceptance. The authors addressed the reviewers' concerns by adding additional experiments (with a 1B LLM) and clarifying their methodology. Overall, the paper is a valuable contribution to the field, offering a practical solution to a common problem in LLM fine-tuning.

**Audience:**

This work would interest individuals in TMLR's audience as it addresses a significant problem in the fine-tuning of language models, which is relevant to many researchers and practitioners working with large language models (LLMs), particularly on their performance and reliability.

**Claims And Evidence:**

The claims made in the submission are supported by accurate, convincing, and clear evidence. The authors provide both theoretical motivation and empirical results to back up their proposed method, Targeted Negative Training (TNT). They demonstrate its effectiveness in reducing unwanted outputs while maintaining the original model's behavior.